# Semantic code clone detection using hybrid intermediate representations and BiLSTM networks

M. Shahbaz Ismail[1] , Sara Shahzad 📷[1] , Fahmi H. Quradaa 📷[2] *

1 Department of Computer Science, University of Peshawar, Peshawar, Pakistan, 2 Department of Computer Science, Aden Community College, Aden, Yemen

๏ These authors contributed equally to this work.

* Qurada@uop.edu.pk

## Abstract

Semantic code clone detection plays an essential role in software maintenance and quality assurance, as it helps uncover fragments of code that express the same logic even when their syntax has been altered or deliberately obfuscated. In this study, we propose a framework that combines hybrid representation learning with deep bidirectional LSTM networks. The model is applied to two intermediate forms of Java programs—Baf and Jimple—extracted through the Soot framework, which together provide both syntactic structure and semantic detail. This design allows the method to cope with difficult obfuscation strategies such as polymorphism and metamorphism. In our experiments, the framework showed strong and stable performance. Training accuracy reached about 98%, while validation accuracy stayed above 95%, with good generalization across the different clone categories described in the Twilight-Zone taxonomy. When compared with other recurrent models, the BiLSTM consistently performed better, especially when combined with multiple intermediate representations and attention mechanisms. On the BigCloneBench dataset, the approach matched or exceeded the results of state-of-the-art tools, achieving recall and F1-scores of up to 97% on challenging clone types. These findings confirm the practical applicability of hybrid intermediate representations for semantic clone detection and suggest promising directions for future research using transformer-based models and large-scale deployment.

## 1 Introduction

Code clones, or duplicated portions of source code, refer to fragments that carry out the same or very similar functionality even though their syntax may differ. They are a common occurrence in software projects, often arising from copy-and-paste practices or from independent implementations of equivalent logic by different developers. While seemingly harmless at first, clones pose serious challenges for long-term

**Data availability statement:** The dataset is available at:

https://github.com/clonebench/BigCloneBench?tab=readme-ov-file.

**Funding:** The author(s) received no specific funding for this work.

**Competing interests:** The authors have declared that no competing interests exist.

software maintenance, increasing both the cost and effort of evolution. When modifications are needed, every clone instance must be located and updated consistently, which raises the risk of oversight. Moreover, errors embedded in one fragment are frequently propagated across the system through duplication, amplifying their impact. The absence of proper documentation around many clones adds another layer of difficulty, making their detection and management even more complex [1,2].

Researchers classify code clones into four main types based on their structural characteristics and degree of similarity [3]. Type-1 clones are identical fragments, differing only in insignificant aspects such as spacing, formatting, or comments. Type-2 clones maintain structural similarity but may vary in identifiers, literals, or data types. Clones of Type-3 involve more significant changes, including added, removed, or modified statements, while still preserving overall functionality. Type-4 clones, in contrast, can differ widely in syntax but exhibit equivalent behavior or semantics. Between Type-3 and Type-4 lies the so-called 'Twilight Zone' [4], where clones are further categorized according to their syntactic resemblance into Very Strongly Type-3 (VST-3), Strongly Type-3 (ST-3), Moderately Type-3 (MT-3), and Weakly Type-3/Type-4 (WT-3/4).

The identification and management of clones is an essential component of software maintenance and ongoing system evolution. Although prior research has emphasized several drawbacks of code cloning [5,6], it is important to recognize that cloning is not always harmful [7]. Some proponents highlight its practicality and perceived reliability in speeding up development cycles and lowering effort. Nevertheless, automated detection of similarities in code remains highly valuable for various purposes such as detecting malware [8,9], improving program understanding [10], finding refactoring opportunities [11], and exposing copyright violations [12]. Clone detection approaches employ different representation strategies, including token-based, text-based, graph-based, tree-based, and metrics-based methods [13]. Most methods focus on detecting specific clone types, typically up to Type-3. In contrast, detecting pure Type-4 clones remains an understudied challenge, largely due to the difficulty of analyzing true behavioral similarities—posing a major obstacle in the field.

The way source code is represented fundamentally shapes the scope of detectable features and influences the detector's architecture, ultimately determining its efficacy [14]. Contemporary techniques predominantly rely on abstract, high-level representations while paying limited attention to compiled or intermediate forms (such as bytecode or assembly ) that preserve execution semantics. Intriguingly, syntactically divergent code implementing identical functionality often yields analogous intermediate representations (IRs) [15], revealing semantically equivalent clones that high-level analysis might overlook. This relationship between representation choice and semantic clone detection capability [16]. highlights the need for a more nuanced approach. An ideal representation would holistically encode both structural and functional attributes, enabling more accurate identification of sophisticated clones. Consequently, advancing clone detection necessitates innovative representation strategies that systematically capture syntactic patterns and underlying semantics.

Research by Hindle et al. [17] demonstrates that programming languages (PLs) exhibit rich statistical patterns, much like natural languages, which play a crucial

role in program analysis. While these patterns are difficult for humans to discern manually, they highlight the value of machine learning (ML) in processing code. Although various ML approaches have been applied to clone detection, their effectiveness in recognizing semantic clones remains limited, primarily because they rely on manually designed features that are often task-specific [18]. Moreover, studies in ML suggest that human-crafted features may fail to fully represent underlying data patterns, sometimes under performing in contrast to features acquired automatically. Current advances in deep learning (DL) have transformed fields such as natural language processing [19,20] and computer vision [21], and these techniques—especially recurrent neural networks (RNNs)—are now being leveraged for clone detection. DL models can autonomously derive meaningful code features from combined semantic and syntactic representations, terminating the need for predefined distance metrics [22,23]. Most DL-based approaches focus on either high-level [24–26] or low-level representations such as bytecode and intermediate code [27,28], but only a limited number of studies explore hybrid methods that integrate both to extract richer feature sets [29–31]. This paper introduces a DL–based code representation designed to capture rich features of source code. The approach leverages two low-level abstract compiled IRs produced by the Soot framework, namely Jimple and Baf. A Siamese neural network, consisting of two identical subnetworks with BiLSTM units, embeddings, and attention mechanisms, is employed to learn from these representations and capture latent semantic and syntactic patterns within code structures. The key contributions of this research are outlined as follows:

- This study is the first to introduce a novel code representation that combines two low-level abstract compiled code forms *Baf and Jimple* to effectively encode both semantic and syntactic characteristics of Java source code.
- A Siamese Bi-LSTM–based DL model is proposed, which captures underlying source code representations to identify syntactic and semantic clones in Java programs.
- A series of systematic experiments were performed using the BigCloneBench dataset [32], a standard benchmark comprising precisely annotated clone pairs. These experiments were designed to rigorously validate the performance of the proposed framework and to provide a comparative analysis against leading clone detection approaches, with emphasis placed on recall and F1-score metrics.

The current study advances prior research by the authors, where semantic code clone detection was performed using manually engineered features extracted from high- and low-level abstract compiled code representations and processed through machine learning classifiers [33]. In contrast, the present work aims to investigates how RNNs can autonomously learn both syntactic and semantic patterns directly from low-level abstract compiled code representations.

This paper is organized as follows. Sect 2 reviews the related work. Sect 3 introduces the background and motivation. Sect 4 describes the proposed methodology, and Sect 5 presents the experimental results on real-world datasets. Sect 6 analyzes these results and discusses their implications. Sect 7 addresses potential threats to validity. Finally, Sect 8 concludes the paper and outlines directions for future research.

## 2 Related work

Semantic code clone detection has become increasingly important as software systems grow more complex and traditional syntactic methods prove insufficient. Recent solutions leverage IRs, DL models, graph structures, and hybrid feature combinations to precisely detect syntactic and semantic code clones across different PLs and execution environments.

To address variations introduced during compilation, numerous techniques operate on binary or IR-level code. For example, Yong et al. [34] introduced BASSET, a scalable framework that utilizes expression trees and a CNN-based ranking model for binary code search. Similarly, IRBinDiff by Shang et al. [35] employs LLVM-IR, contrastive learning, and graph neural networks to uncover semantic similarities despite diverse compilation paths. Khanh-Khoa et al. [36] presented BinShoo, a robust DL solution designed to handle obfuscated and cross-platform code.

Normalization and transformation at the bytecode level have also yielded promising results. Stefan et al. [37] proposed jNorm, which eliminates compiler-specific artifacts in Java bytecode, enabling effective detection of clones and plagiarism without access to source code. Mariam et al. [38] employed program dependence graphs *PDGs* to support fine-grained clone analysis across both source and binary levels, also contributing to vulnerability detection.

DL continues to be a driving force in clone detection. Hung et al. [39] developed BiSim-Inspector, integrating a Bytes2Vec embedding with a CNN-GRU network. Qasem et al. [40] proposed BinFinder, designed to withstand obfuscation and platform variance. Li et al. [41] introduced ALMOND, a zero-shot learning framework that utilizes pre-trained language models for detecting obfuscated code clones.

Graph-based techniques have also shown strong performance. Jiahao et al. [42] developed Tailor, which utilizes Code Property Graphs (CPGs) within a specialized neural network to capture deep functional similarities. Similarly, Gao et al. [43] proposed SIGMADIFF, which integrates PDG-based graph matching to support scalable and semantically aware pseudocode comparison.

Approaches targeting cross-language and binary-to-source mapping have broadened clone detection capabilities. Wang et al. [44] translated C/C++ code to Java bytecode to support Android code analysis, while Ghader et al. [45] applied decompilation and graph comparison to match binaries with reconstructed source code. Davide [46] tackled clone identification in low-level code through scalable decompilation strategies.

In the Java ecosystem, Schäfer et al. [47] adapted StoneDetector for Java bytecode and demonstrated its effectiveness using BigCloneBench. Yang et al. [48] introduced Robin, which applies symbolic execution to detect patched functions, thus minimizing false positives. Weiss et al. [49] examined security vulnerabilities by tracing the reuse of unsafe code from Stack Overflow in Ethereum smart contracts. Other research has explored various IR and bytecode representations. Andre et al. [50] modeled control structures using dominator paths in Java bytecode CFGs. Salim et al. [51] relied on Jimple IR combined with text-based methods, while Roy et al. [52] introduced SeByte, which analyzes bytecode by extracting instructions, method calls, and data types. Caldeira et al. [53] enhanced C code clone detection using NiCad on LLVM virtual assembly, and Yu et al. [28] employed the Smith-Waterman algorithm for bytecode sequence alignment.

More recently, DL models have evolved to incorporate multi-perspective code representations. Dong et al. [54] developed PBCS, using BiLSTM for detecting similarities in student assignments. Wei et al. [55] presented FCCA, a model that fuses token sequences, ASTs, and CFGs through an attention mechanism. Wenhan et al. [25] introduced a modular tree network for more precise semantic interpretation of AST structures, while Ullah et al. [56] developed CrolSSim to assess cross-language similarity using LSTM.

To further enhance robustness, several advanced architectures have been introduced. Wei and Ming [57] proposed CDPU, a Siamese RNN model trained adversarially to distinguish between genuine clones and deceptive non-clones. Yueming et al. [26] designed SCDetector, combining token and graph-level features in a GRU-based Siamese network. Sun et al. [58] introduced VDSimilar, integrating BiLSTM with attention layers, and DeepSim [59] constructed semantic matrices using control and data flows. Dongjiin et al. [28] fused CFGs, identifiers, and bytecode signals via RNNs, and Wan et al. [60] developed SJBCD, using GloVe embeddings and GRUs to detect clones from Java bytecode.

Expanding this research direction, Quradaa et al. [33,61] introduced a machine learning technique and DL framework that integrates high-level representation (e.g., AST) with low-level abstract compiled representations, specifically Jimple and Baf program dependency graphs generated with the Soot framework. In continuation, the current work introduces a novel unified representation that merges Baf and Jimple intermediates into a consolidated semantic representation. These are processed using Siamese deep network contains BiLSTM unites to capture comprehensive syntactic and semantic relationships. This hybrid representation enhances clone detection accuracy and generalizability.

# 3 Research background

In this section, we walk through the core methods behind our study. We start by explaining how we represent code and introduce Soot, the framework we use to generate low level abstract IRs. Then, we dive into our Bi-LSTM-based RNN architecture, which serves as the backbone of our approach. We'll also cover how attention mechanisms and custom embeddings help improve the model's performance. To tie everything together, we end with a concrete example that shows why we designed the system this way.

## 3.1 Code representations

Understanding and representing code effectively is essential for many code analysis tasks, such as detecting clones, identifying vulnerabilities, predicting bugs, summarizing code, and classifying functions [14,16]. Traditional approaches that rely only on high-level syntax or token-based representations often miss the rich semantic and structural information present in source code fragments [62–67]. To overcome these limitations, recent work has focused on creating representations that combine syntactic, semantic, and contextual information [68,69]. These can be derived from sources such as abstract syntax trees (ASTs), control flow graphs (CFGs), program dependency graphs (PDGs), or even low-level IRs.

DL methods, including graph neural networks (GNNs), transformers, and RNNs, have proven effective in using these representations to improve performance across a variety of code understanding tasks. A strong code representation captures both high-level concepts and low-level operational details, allowing models to recognize patterns more accurately. Additionally, combining static analysis results with deep semantic embeddings can enhance both the robustness and interpretability of code models. Ultimately, the aim is to bridge the gap between the code written by humans and its abstract computational behavior. As software systems become increasingly complex, developing reliable and expressive code representations is key to supporting accurate and efficient software comprehension.

## 3.2 The soot analysis framework

The Soot framework is a widely adopted open-source tool for analyzing and transforming Java programs at the bytecode level. Its main strength is the ability to translate Java bytecode into different IRs, each offering a distinct level of abstraction that supports various kinds of analysis and optimization [70]. Among the four IRs that Soot provides—Jimple, Baf, Shimple, and Grimp—this work concentrates on Jimple and Baf, since they complement each other in capturing both higher-level program logic and low-level execution details.

Jimple represents Java code in a simplified, typed three-address form with only 15 statement types [51,71]. This reduction makes the code easier to follow and analyze, particularly for tasks involving control-flow or data-flow reasoning, where a clear semantic structure is important. Baf, on the other hand, stays closer to the original Java bytecode. It uses a stack-based model with about 60 instruction categories, but it trims away unnecessary syntactic complexity. This balance allows Baf to preserve the essential low-level semantics while still being easier to process than raw bytecode. Together, Jimple and Baf provide complementary perspectives that make them especially useful for program analysis in this study.

## 3.3 Recurrent network (RNN)

Recurrent Neural Networks have been widely applied in many areas of machine learning, including recent work on code clone detection. Broadly, these models fall into two groups: the traditional single-directional RNNs and their bidirectional counterparts [23]. The first group includes basic RNNs as well as more advanced units such as Long Short-Term Memory (LSTM) and Gated Recurrent Units (GRU). Bidirectional models, by contrast, run two RNNs in parallel so that sequences are processed both forwards and backwards, giving rise to architectures like Bi-LSTM, Bi-RNN, and Bi-GRU.

Although standard RNNs can handle input sequences of arbitrary length, they often struggle with exploding or vanishing gradients [72]. To mitigate this, Schuster and Paliwal [73] introduced bidirectional RNNs, which allow the model to take advantage of both past and future context. The LSTM design further addresses gradient instability through memory cells

and gating mechanisms [74], while Bi-LSTMs extend this idea by pairing a forward and backward LSTM to strengthen contextual understanding [75,76]. GRUs, meanwhile, offer a leaner alternative to LSTMs. By replacing the three gates with just two—reset and update—they capture long-term dependencies without the full complexity of an LSTM [77,78]. Similar to LSTMs, GRUs also have a bidirectional variant (Bi-GRU), which improves contextual representation and often trains more efficiently.

In this work, we employ the Bi-LSTM model [75]. The architecture consists of two LSTM layers: one processes the input sequence from left to right, while the other works in the opposite direction. For a given identifier sequence $x = [x_1, \rightarrow, x_T]$, the forward LSTM generates a hidden state $h_t^f$ for each time step as it moves from t=1 to T. At the same time, the backward LSTM compiles the sequence in reverse, generating backward hidden states $h_t^b$. The final representation at each step is formed by combining the forward and backward states into a merged hidden vector $h_t^{merged}$, which captures information from both directions of the sequence.

## 3.4 Mechanism of attention

The attention mechanism, first introduced by Bahdanau et al. [79], allows the model to dynamically focus on different parts of the input sequence when generating an output. We employ an additive attention scoring function to compute the relevance of each encoder hidden state to the current decoder step [80].

Let $H = (h^{(1)}, \rightarrow, h^{(T)})$ represent the sequence of hidden states from the encoder. For each decoder state $s_{t-1}$, we compute an alignment score $e_{t,i}$ for every encoder state $h_i$:

$$e_{t,i} = V_a^T tanh(W_a S_{t-1} + U_a h_i) \tag{1}$$

Where $W_a, U_a, and V_a$ are learnable weight matrices and vector. These scores are normalized into attention weights $\alpha_{t,i}$ using a softmax function:

$$\alpha_{t,i} = \frac{\exp(e_{t,i})}{\sum_{j=1}^{T} \exp(e_{t,j})} \tag{2}$$

The context vector $C_t$, which captures the relevant input information for time step $t$, is then computed as the weighted sum of the encoder hidden states:

$$c_x = \sum_{i=1}^{T} (\alpha_{t,i} h_i) \tag{3}$$

This context vector is subsequently used by the decoder to generate the output.

## 3.5 Word embedding (Word2Vec)

Word2Vec is a neural embedding technique that represents words as vectors in a continuous space, where their position reflects the context in which they occur [81]. The approach, introduced by Mikolov et al. [82] at Google, became popular because of its simplicity and effectiveness. It relies on two alternative training strategies: Continuous Bag of Words (CBOW), which learns to predict a word from its surrounding context, and Skip-Gram, which works in the opposite direction by predicting the context given a target word. What makes Word2Vec particularly useful is its ability to capture subtle semantic relationships, for example, vector arithmetic like *king–man + woman ≈ queen* works surprisingly well. Integrating word embedding techniques into source code analysis is a widely adopted approach, owing to the structural parallels between PLs and natural language. In this study, we begin by transforming source code fragments into two abstract IRs, Baf and Jimple, using the Soot framework. From these representations, we extract Baf instructions and Jimple statements

separately. To capture the contextual semantics of each representation, we train two independent Word2Vec models: one to embed Baf instructions and the other for Jimple statements. These models convert the extracted code elements into dense vector representations, enabling downstream neural models to better learn and reason about both syntactic structure and semantic meaning.

## 3.6 Research motivation

Static analysis tools, such as the Soot Framework, facilitate deeper source code comprehension by generating IRs at varying abstraction levels. These IRs standardize syntactically diverse but semantically similar code structures, reducing variability and simplifying analysis [53,83]. The Soot Framework further optimizes IRs by eliminating redundant elements, including unused variables, injected code, and common sub-expressions. Additionally, it normalizes control flow structures (such as for and while loops) into a uniform IR representation (e.g., *goto* instructions), enhancing code homogeneity [51].

To illustrate the necessity of IR-based analysis, consider the two functions in Fig 1(a), which perform similar arithmetic operations, one adding integers and the other adding floating-point numbers. When compiled into Java bytecode (e.g., using *javac*) and disassembled (via *javap*), their bytecode instructions exhibit significant differences Fig 1(b). This divergence arises from Java bytecode's instruction set, which includes specialized operations like *iadd* (integer addition) and *fadd* (floating-point addition). With over 250 distinct bytecode instructions [84], detecting semantic similarities at this level is inherently challenging.

To overcome this limitation, Soot generates Baf IR, a reduced bytecode representation with approximately 60 instructions As shown in Fig 1(c), Baf IR significantly reduces syntactic disparities between functionally equivalent code fragments. By operating at this higher abstraction level, clone detection techniques can more effectively identify semantic clones, code segments that perform the same logic despite structural differences.

```
Public static int Integer_Sum(){
    int x = 80;
    x= x + 20;
    return x;   }
```

```
Public static float Float_Sum(){
    float f = 80.0;
    f= f + 20.0;
    return f;   }
```

(a) Two functions for summation using integer and float data types.

```
Public static int Integer_Sum();
        Code:
        0: bipush    80
        2: istor_0
        3: iload_0
        4: bipush    20
        6: iadd
        7: istore_0
        8: iload_0
        9: ireturn
```

```
Public static float Float_Sum();
        Code:
        0: ldc       #3//flaot 80.0f
        2: fstor_0
        3: fload_0
        4: ldc       #4// float 20.0f
        6: fadd
        7: fstore_0
        8: fload_0
        9: freturn
```

(b) Bytecode of the functions Integer_Sum() and float_Sum().

```
    Code:
        push 80;
        push 20;
        add;
        return;
```

```
    Code:
        push 80.0F;
        push 20.0F;
        add;
        return;
```

(c) Baf IR of the functions Integer_Sum() and float_Sum().

**Fig 1**. **Representation of two Java functions in both bytecode and Baf representations.**

Soot further refines code analysis through Jimple IR, a highly optimized IR that removes redundancies (e.g., dead code, unused variables) and simplifies expressions into three-address code (maximum three operands per statement). Fig 2(a) demonstrates this optimization, where extraneous variables and dead code are eliminated, leaving only the semantically essential components. Jimple IR also standardizes control flow constructs—converting loops (e.g., *for, while*) into if and goto constructs Fig 2(b). This normalization minimizes syntactic variability, improving the detection of functionally equivalent code. Crucially, these optimizations hinder obfuscation techniques (e.g., metamorphism, polymorphism [85]), which rely on syntactic alterations while preserving semantics.

Utilizing static analysis techniques such as the Soot framework, developers and researchers can convert source code into optimized IRs (Baf and Jimple), greatly improving code consistency and facilitating clone detection. The elimination of redundancies and abstract IRs improve code comprehension while mitigating the effects of obfuscation. These advancements not only streamline code analysis but also strengthen the robustness of software engineering tools against adversarial modifications.

(a) Code fragment validate () and its optimized Jimple IR.

(b) Code fragments with different loop statement and their Jimple IR.

**Fig 2. Jimple representations for two Java code fragments.**

# 4 Proposed approach

Although multiple machine learning (ML) methods exist, this research uses DL, particularly RNNs, to analyze code representations from different IRs for identifying semantic clone pairs. RNNs, a type of DL model, were chosen because their multi-layered neuron connections enable automated feature learning. The following section discusses the methodology along with the procedures used for training and testing.

## 4.1 Methodology

As shown in Fig 3, the proposed approach consists of three main steps: (1) preprocessing and transforming the code, (2) using a DL model to learn code representations, and (3) applying a comparator network.

**4.1.1 Preprocessing and IRs generation stage.** In preprocessing, the Stubbler tool [29] is first employed to normalize and compile Java source files without requiring external dependencies. This process produces a normalized Java file together with its corresponding compiled JAR artifact. From these artifacts, individual code fragments are extracted for analysis. To capture richer syntactic and semantic characteristics, the Soot Framework [70] is then applied to the JAR files. For each code fragment, Soot generates two enhanced intermediate representations: (i) the Baf IR, which is transformed into a sequence of abstract bytecode instructions denoted as $Baf_{seq} = Instruction_1, Instruction_2, ..., Instruction_n$ and (ii) the Jimple IR, represented as a sequence of statements $Jimple_{seq} = Statment_1, Statment_2, ..., Statment_n$. Together, these two sequences constitute the main outputs of the preprocessing stage, providing a complementary reflection of the structural and semantic aspects of the code fragments. Further details of this procedure can be found in our earlier work [33,61] and refer to the supporting file S1.

**4.1.2 Code representations of syntactic and semantic characteristics.** Our approach to code similarity centers on a Siamese network design [86], chosen for its proven ability to handle comparative tasks. The core of this architecture

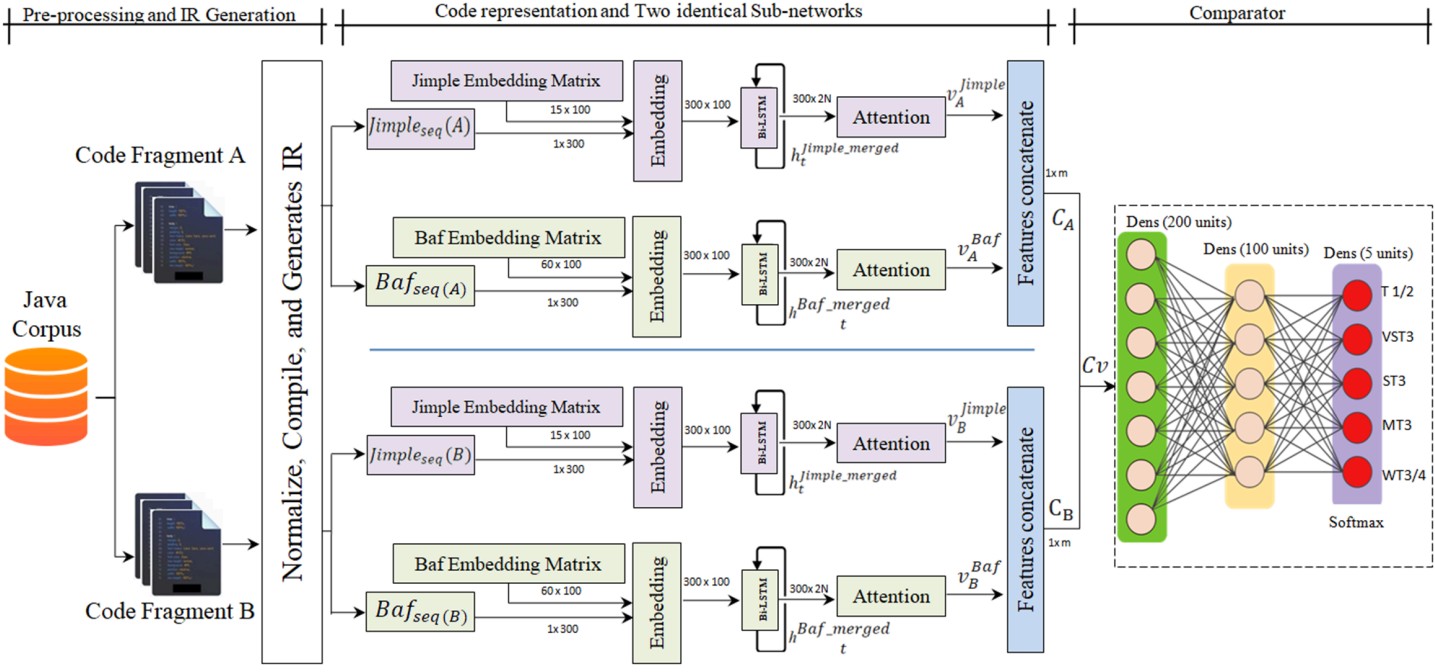

**Fig 3**. Architecture of the proposed technique.

is its twin networks with tied weights, which force the model to learn a unified representation space for both code fragments. This shared-weight design was crucial for us, as it reduced computational overhead and prevented the model from developing inconsistent feature extractors for each input [87,88].

The processing pipeline, detailed in Fig 3, works on pairs of code fragments. We represent each fragment in two ways: as a Jimple statement sequence $Jimple_{seq}$ to get a clearer Syntactic picture and as a Baf instruction sequence $Baf_{seq}$ to retain syntactic details. These sequences are then fed into the parallel subnetworks. The first step for each is an embedding layer, which projects the code tokens into a continuous vector space. These embeddings are then passed through a bidirectional LSTM to capture long-range dependencies in the code. Finally, we found that a standard Bi-LSTM wasn't enough; it tended to weigh all statements equally. To address this, we added an attention layer to allow the model to dynamically highlight the most informative parts of the code sequence.

- **Token-Based Representations**

    We model the sequential nature of the $Jimple_{seq}$ and $Baf_{seq}$ token streams using two separate bidirectional LSTM (Bi-LSTM) networks [75]. The bidirectional processing capability is essential for code understanding, as the semantic meaning of a token (e.g., a variable name or operator) is highly dependent on both its preceding and subsequent context (As illustrated in Fig 4).

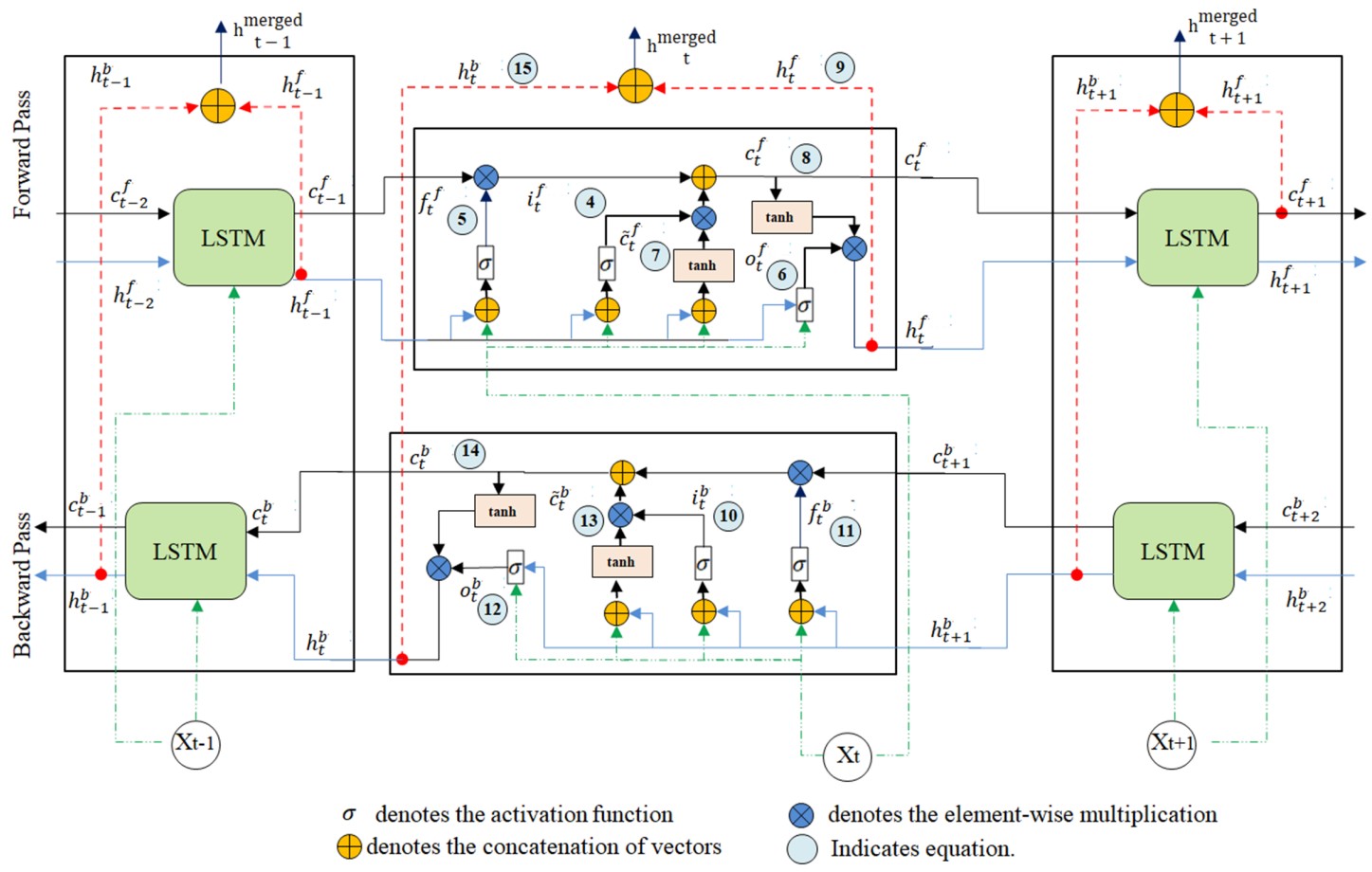

**Fig 4**. Detailed internal architecture of a Bi-LSTM.

As shown in Fig 4, each Bi-LSTM processes a sequence through three stages:

(i) a forward pass responsible for encoding left-hand contextual dependencies, generating the contextual representation $h_t^f$, as specified by Equations 4-9, Input gate:

$$i_t^f = \sigma(W_i^f x_t + U_i^f h_{t-1}^f + b_i^f) \tag{4}$$

Forget gate:

$$f_t^f = \sigma(W_f^f x_t + U_f^f h_{t-1}^f + b_f^f) \tag{5}$$

Output gate:

$$o_t^f = \sigma(W_o^f x_t + U_o^f h_{t-1}^f + b_o^f) \tag{6}$$

Candidate cell state:

$$\tilde{C}_t^f = \tanh(W_c^f x_t + U_c^f h_{t-1}^f + b_c^f) \tag{7}$$

Cell state update:

$$c_t^f = (f_t^f \odot C_{t-1}^f) + (i_t^f \odot \tilde{C}_t^f) \tag{8}$$

Forward hidden state output:

$$h_t^f = o_t^f \odot \tanh(c_t^f) \tag{9}$$

Here $f_t^f, i_t^f,$ and $o_t^f$ are the forget gate (decides what to discard from previous state), input gate (controls new information to store), and output gate (determines the next hidden state) for forward pass respectively. $\tilde{C}_t^f$ is the potential update (Candidate cell state), $c_t^f$ is the current cell state (memory of the LSTM) at time $t$, and $h_t^f$ is the forward contextual representation at time $t$. $W_*^f$, and $U_*^f$ are weight matrices and $b_*^f$ are bias vectors. $\sigma$ is the Sigmoid activation function and $\odot$ donates Element-wise multiplication.

(ii) A backward pass responsible for encoding the right-hand contextual dependencies, $h_t^b$ as defined in Equations 10-15. Input gate:

$$i_t^b = \sigma(W_i^b x_t + U_i^b h_{t+1}^b + b_i^b) \tag{10}$$

Forget gate:

$$f_t^b = \sigma(W_f^b x_t + U_f^b h_{t+1}^b + b_f^b) \tag{11}$$

Output gate:

$$o_t^b = \sigma(W_o^b x_t + U_o^b h_{t+1}^b + b_o^b) \tag{12}$$

Candidate cell state:

$$\tilde{C}_t^b = \tanh(W_c^b x_t + U_c^b h_{t+1}^b + b_c^b) \tag{13}$$

Cell state update:

$$c_t^b = (f_t^b \odot C_{t+1}^b) + (i_t^b \odot \tilde{C}_t^b) \tag{14}$$

Forward hidden state output:

$$h_t^b = o_t^b \odot \tanh(c_t^b) \tag{15}$$

Here $f_t^b, i_t^b$, and $o_t^b$ donate the gates for the backward pass, $\tilde{C}_t^b$ the candidate cell state, $c_t^b$ refers to the cell state, $h_t^b$ is the backward contextual representation at time $t$. $W_*^b, U_*^f$, and $b_*^b$ are learnable parameters. $\sigma$ is the Sigmoid activation function and $\odot$ donates Element-wise multiplication.

(iii) A fusion of the two resulting hidden-state representations.

The final contextual representation for each token at position $t$ is given by $h_t$, which is formed by concatenating the corresponding forward and backward hidden states, ($h_t^f$ and $h_t^b$), as shown in Equation 16.

$$h_t = Concat[(h_t^f, (h_t^b] \tag{16}$$

This vector $h_t$ encapsulates information from the entire sequence and serves as the input to subsequent layers. The internal mechanics of the LSTM cells, which update their state through input, forget, and output gates, follow the standard implementation defined in [75].

The outputs of the two independent Bi-LSTM unites are the final encoded sequences for each IR, which we denote as $h^{Jimple\_merged}$ and $h^{Baf\_merged}$.

- **Embedding of Jimple and Baf Instructions**

    The proposed model converts code fragments as sequences of Jimple statements, $Jimple_{seq}$, and Baf instructions, $Baf_{seq}$. To make these sequences suitable for DL models similar to RNNs, two Word2Vec embeddings were trained separately for Jimple and Baf tokens. To create the training dataset, over 41,850 files were processed: functions were captured and subsequently converted into Baf and Jimple IRs with the Soot framework. Then, Jimple statements and Baf instructions were then converted into sequences and stored in a database for model training.

    In total, 515,654 functions were processed, providing ample data to train both Word2Vec models. The models produced fixed-length 100-dimensional vector embeddings for each Jimple and Baf identifier. After training, the resulting embedding matrices were derived and integrated into the model's embedding layers (see Fig 3), with one layer assigned to Jimple statements and a separate layer for Baf instructions.

- Attention Mechanism and Feature Integration Following the general attention mechanism described in Sect 3.4, attention layers were incorporated after each Bi-LSTM layer to selectively focus on important segments of the input sequences. When the Jimple statement sequence $Jimple_{seq}$ is processed by the Bi-LSTM unit, the resulting hidden state vector $h_t^{Jimple\_merged}$, represents the sequence's contextual information. The attention mechanism computes the alignment scores, normalizes them via softmax, and generates a weighted context vector as follows:

$$e_t^{Jimple} = \tanh\left(W^{Jimple} h_t^{Jimple\_merged} + b^{Jimple}\right) \tag{17}$$

$$\alpha_t^{Jimple} = \text{softmax}(e_t^{Jimple}) \tag{18}$$

$$v^{Jimple} = \sum_{i=1}^{N}(\alpha_t^{Jimple} \cdot h_t^{Jimple\_merged}) \tag{19}$$

Here, $e_t^{Jimple}$ is the alignment score of $h_t^{Jimple\_merged}$, $W^{Jimple}$ and $b^{Jimple}$ are tranable parameter, $\alpha_t^{Jimple}$ represents the normalized attention weights, and $v_t^{Jimple}$ is the weighted representation vector for the Jimple sequence.

Similarly, for the Baf instruction sequence, $Baf_{Seq}$, the attention mechanism operates on the BiLSTM hidden states $h_t^{Baf\_merged}$ to compute the attention scores, as defined in Equations 20 - 22:

$$e_t^{Baf} = \tanh\left(W^{Baf} h_t^{Baf\_merged} + b^{Baf}\right) \tag{20}$$

$$\alpha_t^{Baf} = \text{softmax}(e_t^{Baf}) \tag{21}$$

$$v^{Baf} = \sum_{i=1}^{N}(\alpha_t^{Baf} \cdot h_t^{Baf\_merged}) \tag{22}$$

The resulting vectors $v^{Jimple}$ and $v^{Baf}$ are then fused through concatenation to form a unified representation, as defined in Equation 23. Concatenation is chosen over alternative fusion strategies such as summation or averaging, as it preserves the full dimensional information from both representations, allowing the model to exploit complementary features captured by the $Jimple_{Seq}$ and $Baf_{Seq}$ sequences.

$$C = concat[v^{Jimple}, v^{Baf}] \tag{23}$$

As shown in Fig 3, the Siamese architecture consists of two sub-network, each processing a separate code fragment. After applying the attention mechanism and feature fusion, each sub-network produces a context vector *(C)* representing its input. For a pair of code fragments, the resulting context vectors $C_A$ and $C_B$ are concatenated (Equation 24) and passed to the subsequent stage for similarity learning:.

$$Cv = concat[C_A, C_B] \tag{24}$$

**4.1.3 Comparator network.** The Siamese network generates an output vector, Cv, which encodes the joint representation of code fragments A and B. This vector is then forwarded to a comparator network composed of two fully connected layers of sizes 200 and 100, followed by a softmax layer with five units, as illustrated in Fig 3. The comparator network evaluates similarity and maps the input into a probability distribution over multiple classes, following these steps:

1. First hidden layer: The input vector undergoes a linear transformation:

$$z^{(1)} = Weights^{(1)}C_v + biases^{(1)} \tag{25}$$

   Followed by a *ReLU* activation:

$$a^{(1)} = ReLU(z^{(1)}) \tag{26}$$

2. Second Hidden Layer: The transformed features are further processed through:

$$z^{(2)} = Weights^{(2)}a^{(1)} + biases^{(2)} \tag{27}$$

   and activated using ReLU:

$$a^{(2)} = ReLU(z^{(2)}) \tag{28}$$

3. Output Layer (Softmax): Finally, the network computes:

$$z^{(3)} = Weights^{(3)}a^{(2)} + biases^{(3)} \tag{29}$$

and applies the softmax function to obtain the probability distribution:

$$a^{(3)} = Softmax(z^{(3)}) \tag{30}$$

Here $Cv$ is the context vector input, $z^i$ represents the linear transformation at layer $i$, $weights^{(i)}$ and $biases^{(i)}$ are the trainable parameters, $ReLU$ serves as the activation function for hidden layers, and softmax normalizes the outputs into class probabilities for prediction.

## 4.2 Deep neural network training

Our approach begins with a dataset D composed of code fragment pairs, each annotated with its corresponding clone type. Each entry in the dataset is of the form $((A_1, B_1), y_1), ((A_2, B_2), y_2), ..., ((A_N, B_N), y_N)$, where $A_i$ and $B_i$ are individual code fragments, and $y_i$ is the clone label for that pair. The possible labels $y_i$ belong to the set $C = C_0, C_1, C_2, C_3, C_4$. In this context, $C_0$ covers Type-1 and Type-2 clones, whereas $C_1, C_2, C_3$, and $C_4$ denote the semantic clone categories, VST-3, ST-3, MT-3, and WT-3/4, as defined in BigCloneBench. Once a code pair $(A_1, B_1)$, is normalized and transformed, two outputs are produced for each fragment, $Baf_{Seq}$ and $Jimple_{Seq}$. These outputs are processed through twin sub-networks organized in a Siamese architecture to extract both semantic and syntactic representations. The resulting context vector $Cv$ encapsulates the combined contextual features of $(A_1, B_1)$ and is subsequently processed through a comparator network to assess their similarity.

As the task is formulated as multi-class classification problem with a Softmax activation in the comparator network's output layer, categorical cross-entropy loss is adopted [89]. The batch is calculated as:

$$L(Y, \tilde{Y}) = -\frac{1}{N} \sum_{i=1}^{N} \sum_{j=1}^{C} y_{i,j} \log(\tilde{y}_{i,j}) \tag{31}$$

Where $Y$ denotes the ground-truth label matrix with dimensions ($batch_size \times num_classes$), and  represents the predicted probability matrix of the same shape. $L(Y, \tilde{Y})$ corresponds to the categorical cross-entropy loss for the batch, where $N$ is the batch size, $C$ is the number of classes. For a given sample $i$ and class $y_{j,j}$ indicates the true label from the set $C_0, C_1, C_2, C_3, C_4$, while $\tilde{y}_{i,j}$ denotes the predicted probability assigned of that class.

# 5 Experimental evaluation

The subsequent sections outline the key components of this study. We begin with an overview of the dataset employed in the experiments, followed by a detailed explanation of the model implementation and its architectural design choices. The experimental results are then reported and discussed to assess the effectiveness of the proposed technique in detecting semantic code clones. This includes a comparative analysis of different RNN-based methods, the influence of combining *Jimple* and *Baf* representations of compiled code, and the role of attention mechanisms within the model. Finally, we benchmark the proposed approach against state-of-the-art methods to evaluate its relative performance in real-world semantic clone detection tasks.

## 5.1 Dataset

The study used the BigCloneBench dataset [32], derived IJaDataset-2.0, containing 55,450 Java files collected from 24,557 open-source projects. Unlike automated clone detectors, human experts manually labeled the data, distinguishing between clones and non-clones based on functional similarities. The most recent release of BigCloneBench includes more than 8.5 million confirmed clone pairs and over 260,000 confirmed false positives, spanning 43 functionalities and

categorized into Type-1 through Type-4 clones. Additionally, clones located between Type-3 and Type-4 are divided into the subtypes VST-3, ST-3, MT-3, and WT-3/4 [4]. For this study, we worked with a subset of this dataset.

For data preparation, Java programs compiled into .class files to generate IRs, specifically Baf and Jimpl. To achieve this, we used the Stubber tool [90], which is capable of compiling Java files even with unresolved dependencies. From the compiled output, 27 distinct functionalities were selected to build the dataset. Using the Soot framework [70], we processed 41,865 Java files containing 515,654 functions, extracting both syntactic and semantic features from *Baf* and *Jimple* IRs. This process yielded 5,968,621 function-level clone pairs and 147,390 non-clone pairs, as labeled by Big-CloneBench. Following standard benchmarking practices [3,4], only clones exceeding minimum thresholds, at least six lines or 50 tokens, were considered.

Table 1 summarizes the dataset with the percentage distribution of each clone type, while further details are available in Supporting File S2.

## 5.2 Details of implementation

This study presents a Siamese neural network model, illustrated in Fig 3, developed to capture both semantic and syntactic features from source code fragments. The design of this model draws inspiration from the architecture proposed in [91]. Each branch of the Siamese network incorporates two embedding layers followed by two Bi-LSTM unites with integrated attention modules, configured to process distinct IRs.

For the first branch, Jimple statements in *Jimple* sequences (denoted as $Jimpl_{Seq}$ are embedded using an embedding layer initialized with a 15×100 matrix. This matrix is pre-trained using a Word2Vec model on a large corpus of *Jimple* sequences. The embedded vectors are passed through a Bi-LSTM layer, which outputs a hidden state representation $h_t^{Jimple\_merged}$. This is then processed by an attention mechanism that produces a context vector, $v^{Jimple}$, capturing the most relevant elements of the *Jimple* input. Similarly, the second embedding layer processes Baf instructions $Baf_{Seq}$ using a 62×100-dimensional embedding matrix, trained via Word2Vec on an extensive Baf instruction dataset. The embedded Baf sequence is input to a second Bi-LSTM layer, generating the hidden representation, $h_t^{baf\_merged}$, which is subsequently refined by another attention mechanism to form the vector, $V^{Baf}$. These two vectors, $V^{Baf}$ and $V^{Jimple}$ are then combined into a unified context vector, $C_A$, representing the overall semantic content of the first fragment A.

The same procedure is applied to the second input fragment B, resulting in the corresponding context vector $C_B$. These two vectors, $C_A$ and $C_B$, are then merged into a single context vector, $C_v$, which encapsulates the joint semantics of both fragments. This fused representation is fed into a comparator module that determines the degree of similarity between the two code fragments.

Weight initialization within the Siamese sub-networks adopts the standard approach of sampling from a small random Gaussian distribution, as described in [92,93]. For training purposes, the dataset is randomly partitioned into an 80% training set and a 20% validation set. The training procedure utilizes the Adam optimizer [94] with a learning rate of 0.001 over the course of 50 epochs. All experiments are executed on Google Colab [95], which offers GPU and TPU support. This cloud-based platform facilitates efficient training and evaluation of DL models, and integrates seamlessly with widely used frameworks such as TensorFlow and PyTorch.

**Table 1**. Dataset information of the dataset.

| Type of clone | Pair Sample Total | Percentage |
|---|---|---|
| Type-1&2 | 23,860 | 0.4% |
| Very Strongly Type-3 (VST-3) | 3,835 | 0.064% |
| Strongly Type-3 (ST-3) | 11,866 | 0.199% |
| Moderate Type-3 (MT-3) | 62,654 | 1.05% |
| Weak Type-3 (WT-3/4) | 5,866,406 | 98.287% |
| Clone Total | 5,968,621 | |

### 5.3 Results and analysis

**5.3.1 Performance evaluation of the proposed code representation in semantic clone detection.** To address the performance of the proposed technique, we conducted a comprehensive empirical study using Baf and Jimple IRs. The model's learning behavior was monitored over 50 epochs, analyzing training and validation accuracy and loss to evaluate its ability to learn meaningful syntactic and semantic patterns and generalize to unseen data. As shown in Fig 5(a), the training accuracy steadily increased from 70.68% to 97.99% by epoch 50, while validation accuracy rose from 80.13% to a peak of 95.45% and stabilized at 94.98% in the final epoch, indicating strong generalization with a minimal gap of 3.01%. Minor fluctuations in validation accuracy between epochs 30–45 likely resulted from stochastic mini-batch sampling, yet accuracy remained consistently above 94% post-epoch 26, reflecting model stability. Fig 5(b) illustrates a corresponding convergence in loss, with training loss decreasing from 0.668 to 0.041 and validation loss from 0.472 to 0.267, despite small perturbations during mid-epochs. Overall, these results demonstrate the model's robust performance in detecting code clones, leveraging the complementary strengths of *Baf* and *Jimple* IRs to capture both syntactic and semantic patterns.

To assess robustness, the framework was tested using the "Twilight-Zone" taxonomy. As depicted in Fig 6, the framework performed exceptionally across all categories, with WT3/4 clones achieving the highest metrics (97% precision, 96% recall, 96% F1-score), highlighting its strength in semantic clone detection. VST3 clones also scored highly (94% precision, 95% recall, 94% F1-score), confirming its effectiveness in identifying structurally similar clones, while ST3 and MT3 clones exhibited slightly lower but still strong performance (89% precision, 91% recall, 90% F1-score and 93% precision, 90% recall, 92% F1-score), suggesting potential areas for fine-tuning. Overall, these results confirm that the proposed Baf-Jimple IR-based framework excels in detecting both syntactic and semantic clones. Its consistent performance across diverse clone types underscores its practical utility for software maintenance and quality assurance, providing a reliable and versatile solution for code clone detection.

**5.3.2 The impact of choosing BiLSTM against other RNN techniques in semantic clone detection using the proposed representation.** To compare BiLSTM with other RNN variants for semantic clone detection, we conducted experiments evaluating both training and validation accuracy. BiLSTM consistently outperformed the other architectures, showing rapid convergence and reaching around 98% training accuracy by the final epochs, as shown in Fig 7. This highlights its ability to effectively capture rich syntactic and semantic patterns in code. BiGRU followed closely with roughly 95% training accuracy, offering an efficient alternative that captures meaningful sequence patterns while using fewer parameters than BiLSTM. In contrast, BiRNN, despite its bidirectional context, achieved about 91% training accuracy, indicating that the absence of gating mechanisms limits its ability to handle long-term dependencies.

Fig 7 show that among traditional RNNs, LSTM and GRU performed better than standard RNNs, with training accuracies around 91% and 90%, respectively, while unmodified RNNs lagged at approximately 87%. These results emphasize the importance of gating mechanisms for modeling long-range dependencies in code sequences.

Validation accuracy trends supported these findings. BiLSTM again led, achieving 95–96% and demonstrating strong generalization on unseen data. BiGRU offered a lighter option with around 91% validation accuracy, showing that minimal trade-offs occur despite reduced complexity. BiRNN reached roughly 88%, suggesting that bidirectional context alone is insufficient without gating. LSTM slightly outperformed GRU in validation, with accuracies of 88% and 87%, while standard RNNs achieved only 85%.

Overall, these results underline the superiority of gated, bidirectional architectures like BiLSTM for semantic clone detection. Simple RNNs, even with bidirectional context, struggle to capture complex semantic relationships, reinforcing the value of gating and sequence modeling in DL for code analysis.

**5.3.3 The influence of integrating Baf and Jimple IRs on semantic clone detection.** To assess the impact of combining Baf and Jimple IRs, we performed a comparative analysis of training and validation performance. The hybrid Baf+Jimple model consistently outperformed models using either representation alone across all training stages.

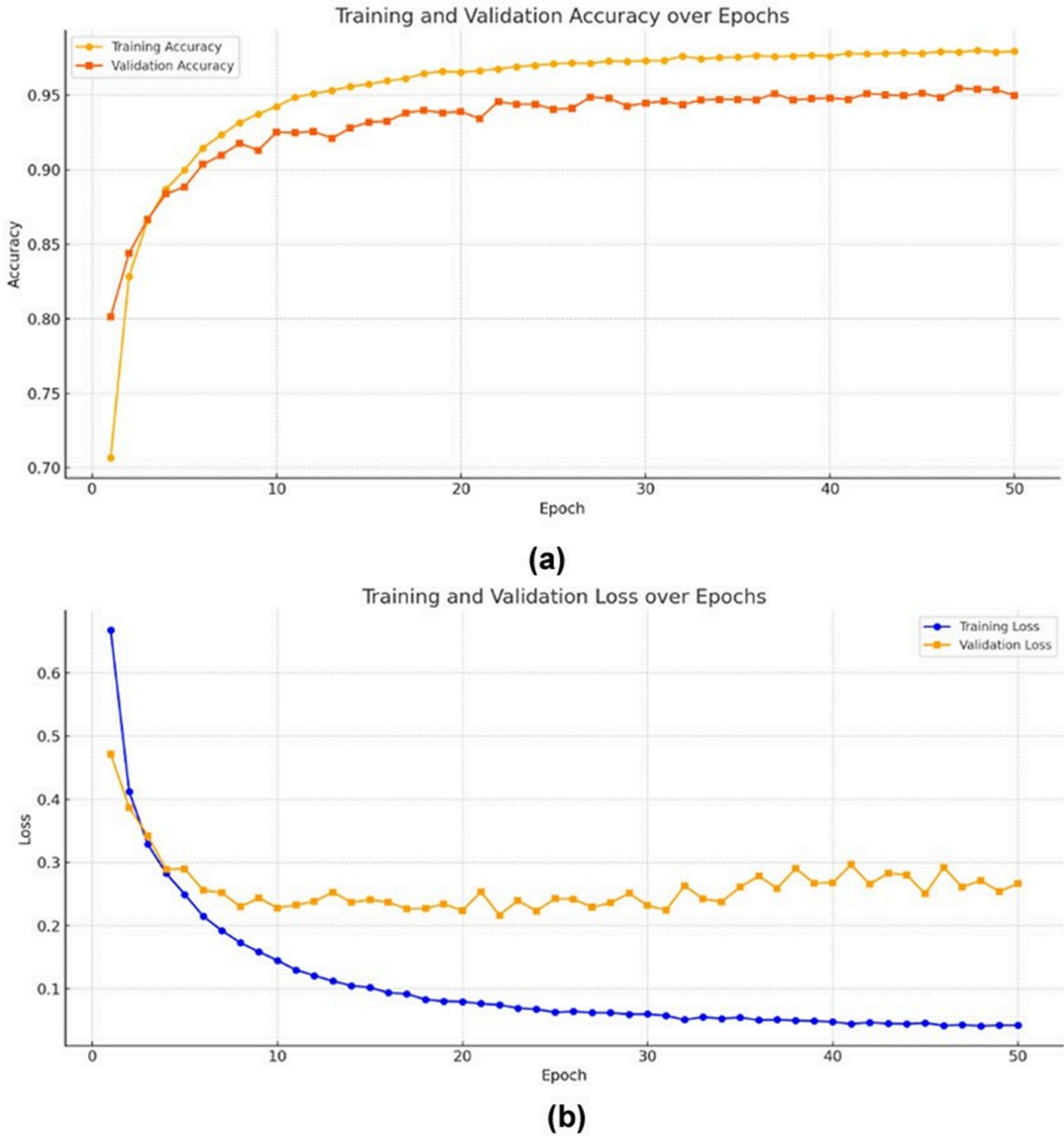

**Fig 5**. **Training versus validation accuracy and loss for the proposed approach.**

As shown in Fig 8, the combined model began with a training accuracy of about 70.7% and steadily increased to nearly 98%, reflecting effective learning and strong convergence. By contrast, the standalone Baf and Jimple models converged

**Fig 6**. Evaluation metrics—recall, precision, and F1-score—for the detected semantic clones.

to final training accuracies of approximately 94% and 93%, underscoring their lower effectiveness relative to the hybrid approach.

Validation results followed a similar trend, with the combined model demonstrating superior generalization. Starting at 80.1% and peaking at roughly 95.3% validation accuracy, it consistently surpassed the individual models. Although the *Jimple*-only model showed a slight advantage over *Baf* in later epochs, both underperformed compared to the integrated framework. This highlights the benefit of merging *Baf's* structural features with Jimple's semantic depth, improving accuracy and generalization while narrowing the training–validation gap and mitigating overfitting.

A brief dip in *Jimple's* training accuracy between epochs 12 and 14 was observed, likely due to mini-batch shuffling or the limited semantic abstraction of *Jimple* representations. However, the model quickly recovered, suggesting this was a temporary optimization fluctuation rather than a structural limitation.

Further evidence of the combined model's effectiveness is seen in the training and validation loss curves. From the outset, Baf+Jimple exhibited a lower initial training loss (0.6680) compared to Baf (0.7607) and Jimple (0.8593). This advantage persisted throughout training, with the hybrid model reaching a final loss of 0.042, clearly outperforming the 0.110 and 0.137 recorded for Baf and Jimple. Validation loss displayed the same trend, with Baf+Jimple consistently maintaining lower values—from 0.4717 down to 0.267—indicating stronger generalization and reduced overfitting.

The superiority of the combined representation is further reflected in classification metrics across all clone categories (Fig 9). In the VST3 category, Baf and Jimple each achieved F1-scores of 93%, while their integration reached 96.50%. For ST3 clones, the standalone models scored 87% (Baf) and 85% (Jimple), whereas the hybrid model attained 92.99%. Similarly, in the MT3 category, the combined model achieved 92.48%, surpassing Baf (88%) and Jimple (86%). The most pronounced improvement appeared in the WT3/4 category, where the hybrid approach achieved an F1-score of 96.50%, compared to 94% and 93% for Baf and Jimple, respectively.

Interestingly, Jimple tends to favor higher recall, while Baf provides a more balanced precision–recall trade-off. By leveraging these complementary strengths, the combined representation delivers enhanced performance across all key metrics, effectively capturing both syntactic structures and deeper semantic relationships in source code.

**5.3.4 The influence of attention layers in the proposed method on overall performance.** Incorporating attention mechanisms into the proposed semantic clone detection model significantly enhances its overall performance [79,96]. By enabling the model to selectively focus on the most relevant parts of the input code, attention layers improve its ability to capture contextual and structural nuances—an essential factor for detecting complex semantic clones.

This advantage is evident across all clone categories, as shown in Fig 10. In the VST3 category, the attention-based model achieved 96% precision, 97% recall, and an F1-score of 96.5%, compared to 94% precision, 92% recall, and a

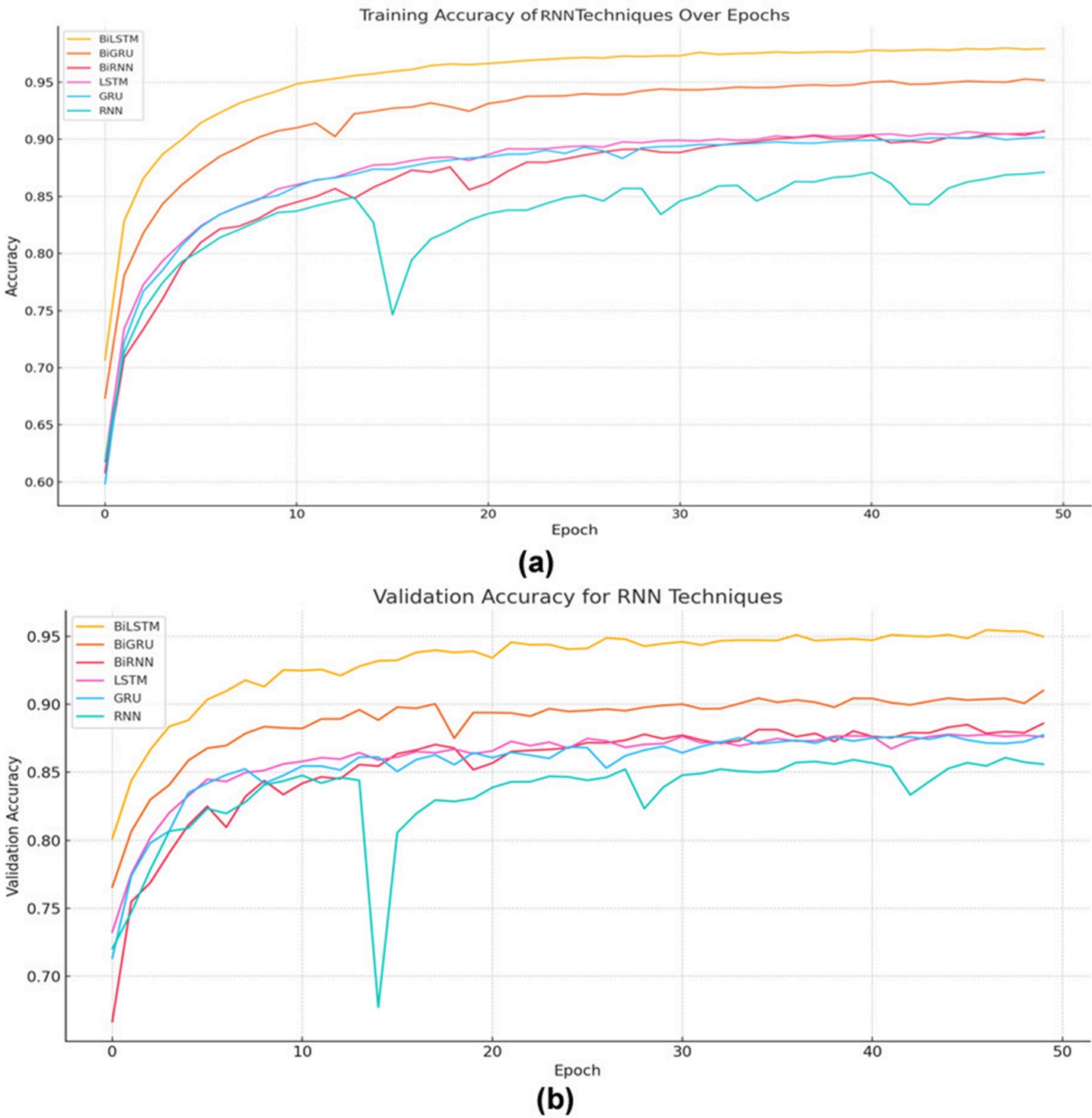

**Fig 7**. **Training and validation accuracy for various RNN methods.**

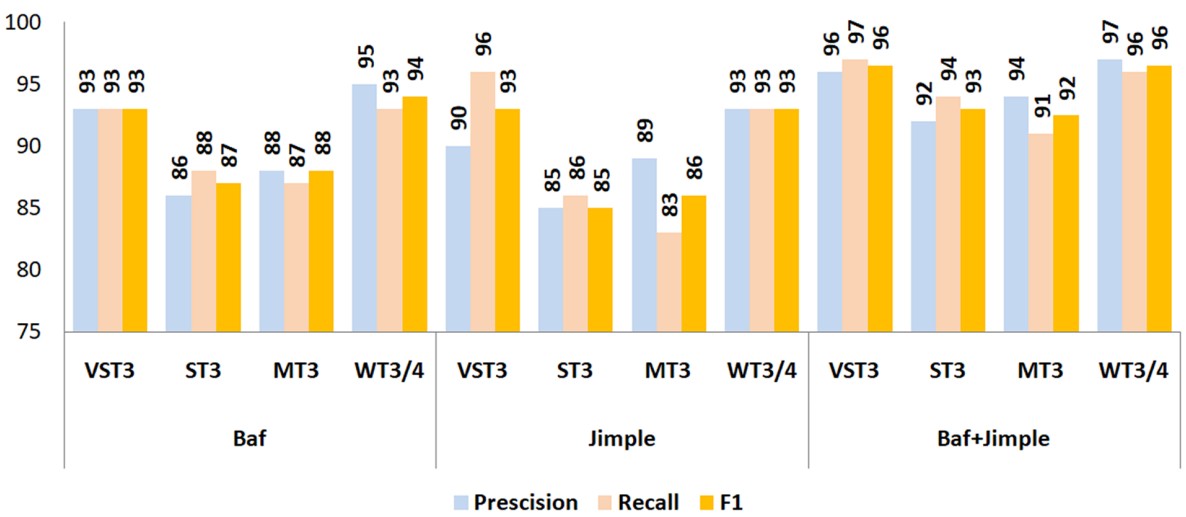

**Fig 8. Performance comparison of *Baf*, *Jimple*, and *Baf+Jimple* in terms of training and validation accuracy and loss.**

**Fig 9. Comparative performance (precision, recall, F1-score) of individual and combined feature representations in semantic clone detection.**

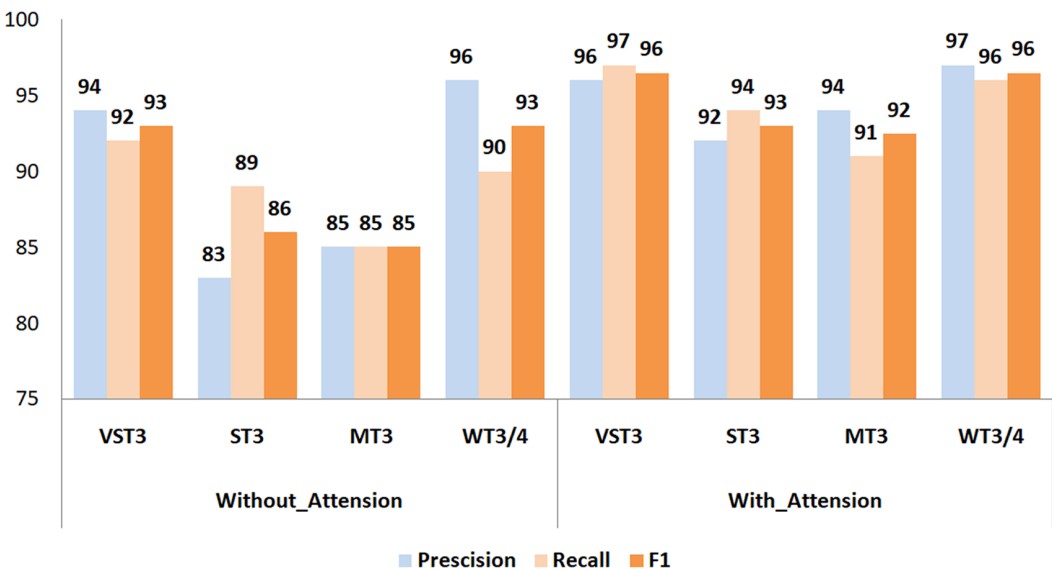

**Fig 10**. Impact of attention layers on precision, recall, and F1-score in semantic clone detection.

93% F1-score for the non-attention variant. Such improvements demonstrate the enhanced discriminative capability afforded by attention mechanisms.

A similar pattern occurs in the ST3 category, where the attention-enhanced model reached 92% precision, 94% recall, and a 92.99% F1-score, whereas the non-attention model lagged with 83% precision, 89% recall, and 86% F1-score, reflecting a limited capacity to detect subtle semantic variations.

The largest gains appear in the more challenging MT3 and WT3/4 categories. For MT3, the F1-score rose from 85% (without attention) to 92.48% (with attention), indicating a marked improvement in identifying semantically rich but structurally diverse clones. Similarly, in the WT3/4 category—widely regarded as the most difficult—the attention-based model achieved an F1-score of 96.5%, surpassing the baseline's 93%.

These results highlight the critical role of attention layers in enhancing the semantic sensitivity of the model. By emphasizing meaningful patterns in the code, attention facilitates more accurate detection of clone relationships that traditional sequence models may overlook. The consistent gains across all categories confirm that attention mechanisms not only boost precision and recall but also reinforce the model's robustness and reliability in real-world clone detection scenarios. Notably, these findings are in close agreement with the results reported in Reference [55].

**5.3.5 Performance comparison of the proposed approach with leading methods for semantic code clone identification.** To thoroughly evaluate the effectiveness of the proposed semantic code clone detection approach, we performed a comparative analysis against several well-established tools in the field. The baseline methods considered include SorcererCC [97], CCFinder [98], NiCad [99], iClones [100], Dekard [101], CCLearner [102], CDLH [103], DLC [27], ASTNN [24], Oreo [4], and the technique introduced by Qurada et al. [33]. For benchmarking, we employed the BigCloneBench dataset, which is widely recognized for its complexity and broad coverage of diverse clone categories. This dataset serves as a standard reference for assessing clone detection techniques under realistic and challenging scenarios.

The evaluation considers four clone categories specified in BigCloneBench: VST3, ST3, MT3, and WT3/4, as Shown in Figs 11 and 12. To provide a fair and reliable comparison, recall and F1-score were selected as the primary performance

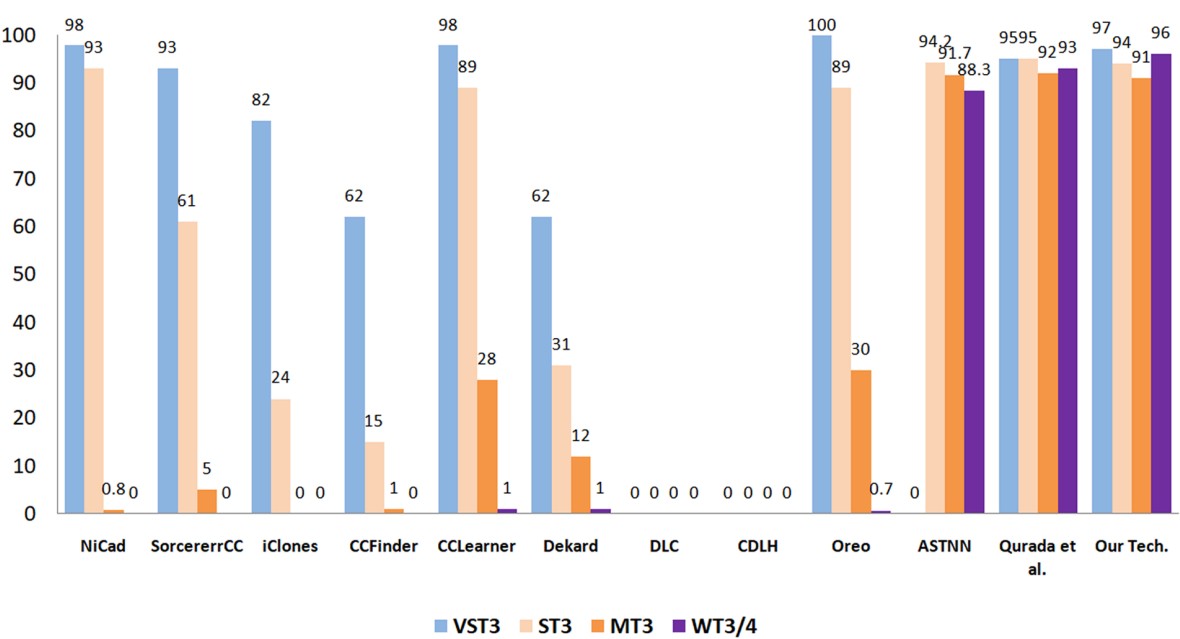

**Fig 11.** Comparing recall of the proposed technique and eleven selected methods across different clone categories.

measures. Recall indicates a tool's effectiveness in retrieving all relevant clones, whereas the F1-score balances precision and recall, offering a more holistic assessment of classification performance.

The proposed approach exhibited strong and competitive results across all clone categories. In the VST3 category, it achieved 97% recall and a 97% F1-score, performing on par with leading tools such as NiCad (98% recall, 98% F1) and Qurada et al. (95% recall, 96% F1). Although Oreo attained 100% recall in this category, it failed to yield a valid F1-score, likely due to precision limitations or incomplete predictions. Within the ST3 category, the proposed method achieved 94% recall and a 93% F1-score, comparable to ASTNN (94.2% recall, 97% F1) and Qurada et al. (95% recall, 95% F1). These findings indicate that the approach is well-suited for capturing strong semantic similarities in code, even in the presence of minor syntactic differences.

The greatest performance improvements are evident in the MT3 and WT3/4 categories, which are recognized for their structural and semantic complexity. In MT3 detection, the proposed approach achieved 91% recall and a 92% F1-score, substantially surpassing classical tools such as NiCad (0.8% recall, 2% F1) and CCFinder (1% recall, 2% F1). Even Oreo, with 30% recall and an invalid F1-score, showed limited effectiveness by comparison. Among existing methods, only ASTNN (91.7% recall, 95.5% F1) and Qurada et al. (92% recall, 91% F1) demonstrated comparable performance in this demanding category.

In the WT3/4 category—characterized by clones with only weak semantic similarity and a high detection difficulty—the proposed method achieved 96% recall and a 96% F1-score. This represents the best performance among all compared techniques and highlights the model's strong ability to capture deeply abstract semantic relationships. The closest competing methods, Qurada et al. (93% recall, 93% F1) and ASTNN (88.3% recall, 93.7% F1), still fell short in both metrics. In contrast, traditional approaches such as NiCad, iClones, and CCFinder completely failed in this category, yielding zero recall and F1.

Taken together, these findings emphasize the robustness and superiority of the proposed approach, particularly in handling semantically abstract and challenging clone types. Its consistently strong performance across all categories, coupled

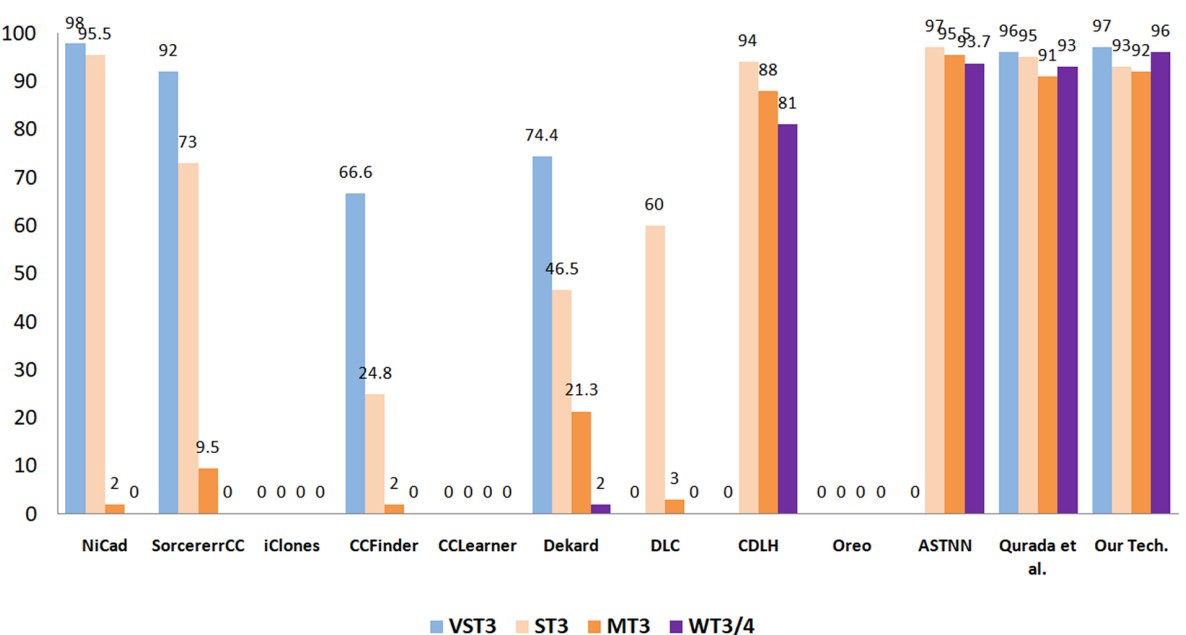

**Fig 12.** Comparing F1-score of the proposed technique and eleven selected methods across different clone categories.

with significant improvements over traditional techniques, demonstrates its practical value for real-world software analysis. The subsequent figures present these comparisons visually, offering a clearer perspective on how each tool performs across the different clone classes.

Present a comparative analysis of recall and F1-scores across various clone types, contrasting the proposed method with established baseline approaches such as Dekard, iClones, SourcererCC, CCFinder, DLC, NiCad, ASTNN, CDLH, Oreo, CCLearner, and Quradaa et al. [33].

## 6 Discussion

Hybrid representation learning (HRL) has emerged as a promising approach in DL because it can bring together multiple types of information into a single, compact feature space. This capability is particularly valuable when dealing with complex data like source code [55,79,104]. Studies have shown that representing code at higher levels of abstraction can significantly improve clone detection, allowing models to capture subtle semantic relationships that simpler representations might miss [59].

A key contribution of this work is the novel application of RNNs for learning code representations directly from abstract IRs. Unlike earlier methods that relied on mixing high- and low-level features extracted from raw code, our approach employs Bi-LSTM networks to learn from Baf and Jimple IRs. This design offers several advantages: it filters out syntactic noise while preserving the essential semantic operations. As a result, the model is more resilient to obfuscation techniques such as polymorphism and metamorphism, which alter code structure without affecting program behavior [85]). The following section provides a detailed analysis of the experimental results, examined from several key perspectives.

**1- Efficacy of the proposed representation and learning stability**

To evaluate the effectiveness of our proposed code representation in capturing syntactic and semantic features, the model was trained for 50 epochs and monitored via training and validation accuracy and loss. Training accuracy increased steadily from 70.68% in the first epoch to 97.99% by epoch 50, while validation accuracy rose from 80.13% to a peak of

95.45% at epoch 48, stabilizing at 94.98%. The narrow 3% gap between training and validation indicates strong generalization and minimal overfitting. Minor fluctuations between epochs 30–45 did not affect overall stability, as validation accuracy remained above 94% after epoch 26. Evaluated across clone types using the Twilight-Zone taxonomy, the model achieved high performance, especially on WT3/4 semantic clones, with recall, precision, and F1-scores all around 96–97%. Strong results were also observed for VST3, ST3, and MT3 clones, though slightly lower, suggesting areas for further refinement. These findings demonstrate that the hybrid Baf–Jimple representation effectively integrates syntactic structure with semantic abstraction, enabling robust detection of semantic and structural clones. This underscores its practical relevance for software maintenance, refactoring, and quality assurance tasks.

### 2- The Impact of Choosing BiLSTM over Other RNN Techniques in Semantic Clone Detection Using the Proposed Representation

We evaluated several RNN variants on their ability to learn semantic code representations from Baf and Jimple IRs. BiLSTM achieved the best performance, with 98% training accuracy and 95–96% validation, likely due to its bidirectional processing and gating mechanisms that capture complex code dependencies. BiGRU followed, reaching 95% training and 91% validation, offering a balance between accuracy and efficiency. BiRNN achieved 91% training and 88% validation, benefiting from bidirectionality but limited by the absence of gates. Non-bidirectional models LSTM and GRU performed moderately well ( 91%/ 90% training; 88%/ 87% validation), while standard RNN showed the weakest results ( 87% training, 85% validation). Overall, these findings confirm that gated and bidirectional architectures—especially BiLSTM—are most effective for semantic clone detection, with BiGRU as a viable alternative in resource-constrained settings.

### 3- Effect of combining Baf and Jimple IRs on identifying semantic clone

We conducted a comparative evaluation to assess the impact of integrating Baf and Jimple IRs. The hybrid model clearly outperformed the individual representations, achieving 97.9% training accuracy and 95.3% validation accuracy, compared to 94%/91% for Baf and 93%/92% for Jimple. Training and validation losses were also reduced (0.042 and 0.267 vs. 0.110/0.137 and 0.4717). F1-scores confirmed these improvements across all clone categories: VST-3 (96.5% vs. 93%), ST-3 (92.99% vs. 87%/85%), MT-3 (92.48% vs. 88%/86%), and WT-3/4 (96.5% vs. 94%/93%). These results indicate that combining Baf's structural information with Jimple's semantic detail strengthens generalization, reduces overfitting, and improves robustness across different clone types.

### 4- Impact of Attention Mechanisms on Model Performance

Integrating attention mechanisms into the proposed semantic clone detection model significantly enhances its capability to detect complex clones. Attention layers allow the model to focus on the most relevant parts of the input, improving its semantic understanding. As shown in Figure 9, the attention-based model outperformed its non-attention counterpart across all clone categories. For VST3 clones, it achieved 96% precision, 97% recall, and a 96.5% F1-score, compared to 94%, 92%, and 93% without attention. In the ST3 category, the F1-score improved from 86% to 92.99%. The largest gains appeared in the MT3 and WT3/4 categories. The F1-score of MT3 rose from 85% to 92.48%, while WT3/4 improved from 93% to 96.5%. These results confirm that attention layers boost the model's recall, precision, and overall robustness, especially in challenging detection scenarios.

### 5- Comparison with State-of-the-Art Approaches

We evaluated our proposed semantic clone detection method against established tools—including NiCad, CCFinder, iClones, SorcererCC, CCLearner, Dekard, DLC, CDLH, Oreo, ASTNN, and Quradaa et al.—using the BigCloneBench dataset. For Very Strong Type-3 (VST3) clones, our approach achieved 97% recall and 97 F1-score, comparable to NiCad (98%, 98%) and Quradaa et al. (95%, 96%), while Oreo had 100% recall but lacked a valid F1. In Strong Type-3 (ST3) clones, we reached 94% recall and 93 F1, close to ASTNN (94.2%, 97%) and Quradaa et al. (95%, 95%). For Moderate Type-3 (MT3), our method achieved 91% recall and 92 F1, significantly outperforming traditional tools like NiCad (0.8%, 2%) and CCFinder (1%, 2%), with only ASTNN (91.7%, 95.5%) and Quradaa et al. (92%, 91) approaching similar results.

In the challenging Weak Type-3/4 (WT3/4) category, our approach obtained the highest 96% recall and 96 F1, surpassing Quradaa et al. (93%, 93%) and ASTNN (88.3%, 93.7$), while traditional tools scored zero. These results confirm the robustness and superiority of our method, particularly in detecting semantically complex clones, highlighting its suitability for practical software maintenance and quality assurance tasks.

## 7 Threats to validity and limitations

Although this study was meticulously designed and rigorously conducted, several validity threats should be acknowledged. The investigation concentrates exclusively on method-level code clones, meaning that certain overlapping clones or those within Java classes may not be identified. This focus is intentional, as prior research indicates that method-level clones are the most common in Java code, ensuring the practical applicability of the proposed approach in real-world software development. The technique was rigorously evaluated on BigCloneBench, a widely recognized benchmark of Java code from real-world repositories. This dataset provides a realistic and dependable evaluation framework, enhancing the reliability of the results. Nevertheless, comparisons with existing baseline methods should be interpreted with caution. Some prior studies, such as those on DLC and CDLH, do not provide comprehensive details on their experimental setups, necessitating reliance on reported results. This may introduce biases or confounding variables that could influence the validity of comparative assessments. Consequently, qualitative comparisons between the proposed method and earlier techniques may be constrained in depth and precision. Additionally, this research is limited to Java, a choice justified by the language's sustained prominence in software development. As evidenced by the TIOBE index, Java remains one of the top five most widely used PLs. Its pivotal role in Android app development further reinforces the relevance and practical impact of this work within contemporary software engineering [105].

## 8 Conclusion

This study highlights the effectiveness of hybrid representation learning, combining abstract IRs (Baf and Jimple) with BiLSTM networks for semantic code clone detection. The proposed model demonstrated strong and stable performance, achieving a training accuracy of 97.99% and a validation accuracy exceeding 95.45%, reflecting robust generalization and minimal overfitting. Notably, integrating Baf and Jimple representations substantially improved detection accuracy and reduced loss compared to using either representation alone, with the hybrid model attaining a training loss of 0.042 and a validation loss of 0.267, surpassing the individual Baf (0.110) and Jimple (0.137) models.

The inclusion of attention mechanisms further improved the model's performance, increasing F1-scores from 85% to 92.48% in the Moderate Type-3 (MT3) category and from 93% to 96.5% in the Weak Type-3/4 (WT3/4) category. Comparative evaluations against state-of-the-art tools on the BigCloneBench dataset showed that our approach achieved recall and F1-scores of 97% and 97% for Very Strong Type-3 (VST3) clones, 94% and 93% for Strong Type-3 (ST3), 91% and 92% for MT3, and 96% and 96% for WT3/4 clones. These results significantly surpass classical tools, many of which scored near zero in the more challenging MT3 and WT3/4 categories.

The incorporation of attention mechanisms further enhanced the model's performance, raising F1-scores from 85% to 92.48% for Moderate Type-3 (MT3) clones and from 93% to 96.5% for Weak Type-3/4 (WT3/4) clones. Comparative evaluations against state-of-the-art tools on the BigCloneBench dataset demonstrated that our approach achieved recall and F1-scores of 97% and 97% for Very Strong Type-3 (VST3) clones, 94% and 93% for Strong Type-3 (ST3), 91% and 92% for MT3, and 96% and 96 for WT3/4 clones. These results markedly outperform classical tools, many of which registered near-zero performance in the more challenging MT3 and WT3/4 categories.

These findings confirm both the superiority and practical relevance of the proposed method for real-world software maintenance and quality assurance, emphasizing the effectiveness of combining DL with hybrid IRs to accurately detect semantically complex code clones.

Future work will focus on investigating transformer-based architectures, including models like BERT and GPT, to enhance semantic clone detection using the integrated code representation. We also aim to adapt and evaluate this approach across various PLs to broaden its applicability and impact. Moreover, although this study did not cover scalability and performance optimization, these areas represent valuable directions for subsequent research efforts.

## Supporting information

**S1 File.** List of *Baf* and *Jimple* features used in this work.
(PDF)

**S2 File.** Descriptive analysis of all clones types in BigCloneBench.
(PDF)

## Acknowledgments

The authors gratefully acknowledge the University of Peshawar for its support in the development of this work.

## Author contributions

**Conceptualization:** M. Shahbaz Ismail, Sara Shahzad, Fahmi H. Quradaa.

**Data curation:** M. Shahbaz Ismail.

**Formal analysis:** M. Shahbaz Ismail, Sara Shahzad, Fahmi H. Quradaa.

**Funding acquisition:** Sara Shahzad.

**Investigation:** Sara Shahzad.

**Methodology:** M. Shahbaz Ismail, Sara Shahzad, Fahmi H. Quradaa.

**Project administration:** Sara Shahzad, Fahmi H. Quradaa.

**Resources:** M. Shahbaz Ismail, Fahmi H. Quradaa.

**Software:** M. Shahbaz Ismail, Sara Shahzad, Fahmi H. Quradaa.

**Supervision:** Sara Shahzad, Fahmi H. Quradaa.

**Validation:** M. Shahbaz Ismail, Sara Shahzad, Fahmi H. Quradaa.

**Visualization:** Fahmi H. Quradaa.

**Writing – original draft:** M. Shahbaz Ismail.

**Writing – review & editing:** M. Shahbaz Ismail, Fahmi H. Quradaa.

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
