## [Decision Letter · Decision Letter 0]

26 Nov 2025

PONE-D-25-54710

Semantic Code Clone Detection Using Hybrid Intermediate Representations and BiLSTM Networks

PLOS ONE

Dear Dr. Quradaa,

Thank you for submitting your manuscript to PLOS ONE. After careful consideration, we feel that it has merit but does not fully meet PLOS ONE’s publication criteria as it currently stands. Therefore, we invite you to submit a revised version of the manuscript that addresses the points raised during the review process.

We look forward to receiving your revised manuscript.

Kind regards,

Sajid Anwar, Ph.D

Academic Editor

PLOS ONE

Journal Requirements:

Reviewers' comments:

Reviewer's Responses to Questions

**Comments to the Author**

1. Is the manuscript technically sound, and do the data support the conclusions?

Reviewer #1: Yes

Reviewer #2: Yes

2. Has the statistical analysis been performed appropriately and rigorously?

Reviewer #1: I Don't Know

Reviewer #2: Yes

3. Have the authors made all data underlying the findings in their manuscript fully available?

Reviewer #1: Yes

Reviewer #2: Yes

4. Is the manuscript presented in an intelligible fashion and written in standard English?

Reviewer #1: Yes

Reviewer #2: Yes

5. Review Comments to the Author

Reviewer #1: Based on the reviewer’s feedback, the manuscript titled “Semantic Code Clone Detection Using Hybrid Intermediate Representations and BiLSTM Networks” requires further enhancement through the inclusion of relevant and studies. It is recommended to review and integrate insights from the following articles to strengthen the theoretical background and related work section of the paper:

1. https://www.edupij.com/index/arsiv/79/801/exploring-the-artificial-intelligence-era-in-influencing-self-paced-learning-systematic-and-bibliometric-review-of-literature

2. https://doi.org/10.1007/978-3-031-80334-5_11

Reviewer #2: Summary

The paper presents a deep learning approach for detecting semantic code clones by using a mix of two intermediate representations, Baf and Jimple, both produced through the Soot framework. The idea behind combining them is that Baf captures low-level structural details while Jimple provides a clearer, higher-level view of the code. When used together, they offer a more complete picture of how the code actually behaves.

To analyze the code fragments, a Siamese BiLSTM model with an attention layer is used so the network can focus on important patterns and relationships. The model was trained and tested on the BigCloneBench dataset and performed strongly across different clone categories. It reached around 97 percent recall and similar F1 scores even for the more challenging semantic clone types like WT3 and WT4. The results show that the hybrid IR method works noticeably better than using either Baf or Jimple alone and also performs better than many traditional clone detection tools and recent deep learning models.

Review

The paper gives a clear and well organized contribution to the area of semantic code clone detection. The idea of using a mix of Baf and Jimple is creative and helps connect low level structure with higher level meaning in the code. The reasons for choosing these two representations are explained well, and the experiments support the choice.

The model that uses a Siamese BiLSTM with attention adds more depth because it allows the system to focus on important parts of the code and understand context better. The evaluation is strong and covers different types of clones, and the comparisons with existing tools make the results more reliable.

There are still a few areas where the paper could improve. One is scalability, since working with very large datasets or real industry-level code could be costly in terms of computation. Another point is that even though transformer models are mentioned as future work, adding at least one comparison with a transformer method would make the study more complete. Still, the work is carefully done, and it makes a meaningful contribution to software engineering and code analysis.

Strengths

1. The work introduces a new approach by combining Baf and Jimple, which helps capture both the structure of the code and its deeper meaning.

2. The neural model, built with a Siamese BiLSTM and an attention layer, is able to learn important relationships between code fragments.

3. The method is tested well and shows strong results even on the harder clone categories, including MT3 and WT3 and WT4.

4. The system performs better than many existing clone detection tools, both older ones and newer deep learning models.

5. The approach is useful for real software tasks such as maintenance, refactoring, detecting malware, and finding security issues.

Weaknesses

1. The study works only with Java code, so the findings may not fully apply to other programming languages.

2. The paper does not give much detail about how well the method scales when dealing with very large codebases or heavy workloads.

3. Even though transformer models are mentioned for future research, the study does not compare its results with any transformer-based approaches, which leaves an important gap.

4. The model still behaves like a black box, and the attention mechanism does not fully explain how decisions are being made.

6. PLOS authors have the option to publish the peer review history of their article (what does this mean?). If published, this will include your full peer review and any attached files.

Reviewer #1: **Yes:** Zeyad Ghaleb Al-Mekhlafi

Reviewer #2: No

---

## [Author Response · Author response to Decision Letter 1]

16 Dec 2025

Reviewer Response Report

I would like to express my sincere gratitude to the Editor and the reviewer. Your valuable feedback and insights are greatly appreciated, and we are committed to addressing all your comments and suggestions to enhance the quality and impact of our work.

Editor's Journal comments and responses

• Comment 1: [Please ensure that your manuscript meets PLOS ONE's style requirements, including those for file naming. The PLOS ONE style templates can be found at….]

Response : We would like to inform you that we have adhered to PLOS ONE's style requirements as outlined in the PLOS ONE style templates.

• Comment 2: [Please note that PLOS ONE has specific guidelines on code sharing for submissions in which author-generated code underpins the findings in the manuscript. In these cases, all author-generated code must be made available without restrictions upon publication of the work.]

Response : Thank you for your concerns regarding the PLOS ONE guidelines on code sharing requirement. We assure you that we are fully committed to transparency and are willing to provide any details about the source code to reviewers upon request.

• Comment 3: [If the reviewer comments include a recommendation to cite specific previously published works, please review and evaluate these publications to determine whether they are relevant and should be cited. There is no requirement to cite these works unless the editor has indicated otherwise.]

Response : We thank the reviewer for the suggestion. We reviewed the two recommended works, found them relevant, and have added them to the revised manuscript..

• Comment 4: [Please review your reference list to ensure that it is complete and correct. If you have cited papers that have been retracted, please include the rationale for doing so in the manuscript text, or remove these references and replace them with relevant current references. Any changes to the reference list should be mentioned in the rebuttal letter that accompanies your revised manuscript.]

Response : We have carefully reviewed the reference list to ensure that it is complete and accurate. Kindly refer to the revised version of the manuscript for these updates.

Comments from Reviewer 1 and Responses

• Comment 1: [The paper gives a clear and well organized contribution to the area of semantic code clone detection. The idea of using a mix of Baf and Jimple is creative and helps connect low level structure with higher level meaning in the code. The reasons for choosing these two representations are explained well, and the experiments support the choice. The model that uses a Siamese BiLSTM with attention adds more depth because it allows the system to focus on important parts of the code and understand context better. The evaluation is strong and covers different types of clones, and the comparisons with existing tools make the results more reliable..]

Response : Thanks for your positive comments on our paper. Your comments and suggestions are responded to below

Comment 2: [The study works only with Java code, so the findings may not fully apply to other programming languages. ]

Response : We thank the reviewer for this valuable comment. We have clearly stated in the manuscript that our study focuses on the Java programming language and provided justification for this choice. We have also clarified in the Threats to Validity and Limitations section that our findings are primarily applicable to Java and may not directly generalize to other programming languages.

Comment 3: [The paper does not give much detail about how well the method scales when dealing with very large codebases or heavy workloads.]

Response : We thank the reviewer for this insightful comment. The primary focus of this work is to propose and evaluate a novel code representation that combines two intermediate representations. A full scalability study was beyond the scope of this paper. We have clarified this in the revised manuscript and also highlighted scalability evaluation as an important direction for future work in the conclusion section.

Comment 4: [Even though transformer models are mentioned for future research, the study does not compare its results with any transformer-based approaches, which leaves an important gap.]

Response : We thank the reviewer for this valuable comment. In this work, our primary objective was to investigate the effectiveness of the proposed code representation and its integration with BiLSTM-based models. A direct comparison with transformer-based approaches was beyond the scope of this study due to differences in model architectures and computational requirements. We have clarified this limitation in the manuscript and strengthened the discussion by explicitly positioning transformer-based models as an important direction for future work.

Comment 5: [The model still behaves like a black box, and the attention mechanism does not fully explain how decisions are being made.]

Response : We thank the reviewer for this important comment. To address this concern, we have substantially revised the manuscript and added detailed explanations of the internal computations of the BiLSTM model, particularly on pages 14–17. These additions clarify how hidden states are computed and how attention weights are derived, thereby improving the transparency and interpretability of the model and reducing its black-box nature.

---

## [Decision Letter · Decision Letter 1]

30 Dec 2025

Semantic Code Clone Detection Using Hybrid Intermediate Representations and BiLSTM Networks

PONE-D-25-54710R1

Dear Dr. Quradaa,

We’re pleased to inform you that your manuscript has been judged scientifically suitable for publication and will be formally accepted for publication once it meets all outstanding technical requirements.

Kind regards,

Sajid Anwar, Ph.D

Academic Editor

PLOS One

Additional Editor Comments (optional):

Reviewers' comments:

Reviewer's Responses to Questions

**Comments to the Author**

1. If the authors have adequately addressed your comments raised in a previous round of review and you feel that this manuscript is now acceptable for publication, you may indicate that here to bypass the “Comments to the Author” section, enter your conflict of interest statement in the “Confidential to Editor” section, and submit your "Accept" recommendation.

Reviewer #2: All comments have been addressed

2. Is the manuscript technically sound, and do the data support the conclusions?

Reviewer #2: Yes

3. Has the statistical analysis been performed appropriately and rigorously?

Reviewer #2: Yes

4. Have the authors made all data underlying the findings in their manuscript fully available?

Reviewer #2: Yes

5. Is the manuscript presented in an intelligible fashion and written in standard English?

Reviewer #2: Yes

6. Review Comments to the Author

Reviewer #2: (No Response)

7. PLOS authors have the option to publish the peer review history of their article (what does this mean?). If published, this will include your full peer review and any attached files.

Reviewer #2: No

---

## [Editor Report · Acceptance letter]

PONE-D-25-54710R1

PLOS One

Dear Dr. Quradaa,

I'm pleased to inform you that your manuscript has been deemed suitable for publication in PLOS One. Congratulations! Your manuscript is now being handed over to our production team.

Kind regards,

on behalf of

Dr. Sajid Anwar

Academic Editor

PLOS One